# Exploring the Underlying Mechanism of Ren-Shen-Bai-Du Powder for Treating Inflammatory Bowel Disease Based on Network Pharmacology and Molecular Docking

**DOI:** 10.3390/ph15091038

**Published:** 2022-08-23

**Authors:** Ni Jin, Yao Liu, Peiyu Xiong, Yiyi Zhang, Jingwen Mo, Xiushen Huang, Yi Zhou

**Affiliations:** 1School of Basic Medical College, Chengdu University of Traditional Chinese Medicine, Chengdu 611137, China; 2School of Laboratory Medicine, Chengdu Medical College, Chengdu 610500, China

**Keywords:** Ren-Shen-Bai-Du Powder (RSBDP), inflammatory bowel disease, network pharmacology, molecular docking, underlying mechanism

## Abstract

Ren-Shen-Bai-Du Powder (RSBDP) is currently used for inflammatory bowel disease (IBD) therapy in China. However, its potential mechanism against IBD remains unknown. In this study, we initially identified potential targets of RSBDP against IBD through network pharmacology analysis and molecular docking. Afterwards, the DSS-induced colitis mice model was employed to assess the effects of RSBDP. The results of network pharmacology indicated that a total of 39 main active ingredients in RSBDP generated 309 pairs of drug-ingredient and ingredient-target correspondences through 115 highly relevant targets of IBD. The primary ingredients (quercetin, kaempferol, luteolin, naringenin, and sitosterol) exerted functions through multiple targets that include CYP1B1, CA4/7, and ESR1/2, etc. GO functional enrichment analysis revealed that the targets related to IBD were significantly enriched in the oxidation-reduction process, protein binding, and cytosol. Per the KEGG pathway analysis, pathways in cancer, adherens junction, and nitrogen metabolism were pivotal in the RSBDP’s treatment of IBD. Additionally, molecular docking demonstrated that a set of active ingredients and their targets displayed good bonding capabilities (e.g., kaempferol and AhR with combined energy < 5 kcal/mol). For the animal experiment, oral RSBDP promoted weight recovery, reduced intestinal inflammation, and decreased serum IL-1, IL-6, and IL-8 concentrations in the DSS + RSBDP group. Meanwhile, oral RSBDP significantly up-regulated the mRNA levels of *CA7*, *CPY1B1*, and *PTPN11*; in particular, the expression level of *CYP1B1* in the DSS + RSBDP group was up-regulated by as high as 9-fold compared to the DSS group. Western blot results indicated that the protein levels of AKR1C1, PI3K, AKT, p-AKT, and Bcl-2 were significantly down-regulated, and Bax was significantly up-regulated in the DSS + RSBDP group. Compared to the DSS and control groups, the Bax/Bcl-2 value in the DSS + RSBDP group increased 4-fold and 8-fold, respectively, which suggested that oral RSBDP promotes apoptosis of intestinal epithelial cells. In short, this study established quercetin, kaempferol, luteolin, naringenin, and sitosterol as the primary key active ingredients of RSBDP that exert synergistic therapeutic effects against IBD through modulating the AhR/CYP1B1 and AKR1C1/PI3K/AKT pathways.

## 1. Introduction

Inflammatory bowel disease (IBD) is characterized by an idiopathic chronic inflammatory state of the gastrointestinal tract and encompasses 2 distinct disease states: ulcerative colitis (UC) and Crohn’s disease (CD) [1]. Since officially naming the disease in 1875, the number of people subjected with IBD has progressively increased each year. Currently, IBD has become a globalized disease, with five million IBD patients globally and a prevalence of 0.5% in some developed countries [2]. Clinically, it is a heterogeneous disease with numerous phenotypes that commonly characterized by abdominal pain, diarrhea, mucus-like pus, blood in the stool, and damage of the mucosal barrier [3,4]. However, there is no official name for IBD in Chinese medicine. Its clinical features can be classified as “dysentery” and “diarrhea” and are mainly induced by damp-heat and a diet that damages the spleen and stomach.

Drugs used frequently to treat IBD tend to be expensive and have high side effects, rendering their consistent application challenging [5]. Conversely, traditional Chinese medicine (TCM) has fewer toxic side effects and is endowed with a variety of biological activities and pharmacological effects, including anti-inflammatory, antibacterial, and immunomodulatory effects with its multi-targeted action characteristics distinctively superior in the treatment and prevention of IBD [6]. The earliest known use of Ren-Shen-Bai-Du Powder (RSBDP) can be traced back to Qian Yi’s “Direct formula for children’s drug syndrome” in the Northern Song Dynasty, which consisted of 12 herbs (weight ratio 2:4:2:3:2:2:2:2:2:2:1:1): *Panax ginseng* (Renshen, RS), *Radix bupleuri* (Chaihu, CH), *Radix Peucedani* (Qianhu, QH), *Rhizoma Ligustic Chuanxiong* (Chuanxiong, CX), *Fructus Aurantii* (Zhike, ZK), *Rhizoma et Radix Notopterygii* (Qianghuo, QH), *Radix Angelicae Pubescentis* (Duhuo, DH), *Poria* (Fuling, FL), *Radix Platycodonis* (Jiegeng, JG), *Radix Glycyrrhizae* (Gancao, GC), *Rhizoma Zingiberis Recens* (Shengjiang, SJ), and *Herba Menthae* (Bohe, BH). RSBDP has also shown notable therapeutic properties against infantile diarrhea and has been demonstrated to improve the intestinal mucosal barrier in UC rats [7]. However, the current situation is different from the use of single-target drugs with a specific mechanism, and the potential mechanism of RSBDP for treating IBD is still unclear, which requires further systematic analysis.

The network pharmacology approach includes the use of systems biology, network analysis, connectivity, and multiple effects [8]. It is in line with the holistic and systemic characteristics of TCM and the principles of diagnosis and treatment, which can elucidate the complex network of interactions between disease-specific genes and compounds in TCM herbal medicines. Their association with one another helps reveal the possible molecular mechanism of TCM prescriptions and provides relevant scientific evidence for clinical research [9]. Therefore, the present study aimed to systematically elucidate the mechanism of RSBDP in the treatment of IBD by analyzing the interactional relationships between drug molecules and IBD-related targets through network pharmacology and molecular docking. We also sought to provide a theoretical basis for clinical research. To achieve our objectives, we implemented the technology roadmap in Figure 1.

## 2. Results

### 2.1. Screening for Key Active Ingredients of RSBDP and Prediction of Important Targets

Primary bioactive ingredients in RSBDP and corresponding ADME information were extracted from the TCMSP data server. In this study, a total of 39 main bioactive ingredients were identified in RSBDP and obtained the drug targets for the corresponding ingredients utilizing the SwissTargetPrediction database (Table 1). After de-duplication and elimination of invalid gene IDs, we intersected targets with those downloaded from the GeneCards and Open Targets databases to acquire 115 potential targets of RSBDP against IBD. Data collation yielded 309 pairs of drug-ingredient and ingredient-target correspondences.

### 2.2. The Construction of the Drug-Ingredient-Target Relationship Network

We imported drug–ingredient–target relationships into the Cytoscape (v3.7.2, Paul Shannon, CA, USA) software (containing 12 drugs, 39 active ingredients, and 115 targets) to construct a drug–ingredient–target network diagram for the treatment of IBD with RSBDP (Figure 2). The dark blue color represents the drug composition of RSBDP, the light purple color denotes the main active ingredients of the drug, and the yellow color signifies the common targets of the active ingredients and IBD-related targets. The size of the ingredient node correlates positively with the degrees.

We performed topological analyses of the drug-ingredient-target network in the Cytoscape 3.7.2 software using “degree value” as a screening parameter for the active ingredients. Our results suggest that quercetin, kaempferol, luteolin, naringenin, and sitosterol were the potential key molecules for RSBDP’s treatment of IBD (Table 2).

We also analyzed the core targets of RSBDP for IBD treatment in the same way in the Cytoscape 3.7.2 software. The top 10 targets, namely cytochrome P4501B1 (CYP1B1), carbonic anhydrase 7 (CA7), cytochrome P45019A1 (CYP19A1), carbonic anhydrase 4 (CA4), protein tyrosine phosphatase 1 (PTPN1), estrogen receptor 2 (ESR2), multidrug resistance-associated protein 1 (ABCC1), ATP-binding transporter protein G family member 2 (ABCG2), estrogen receptor 2 (ESR1), and cyclin-dependent kinase 1 (CDK1), are also listed in Table 3.

### 2.3. The Construction and Topological Analysis of the PPI Network

This study imported the 115 obtained common targets into the SRING database to create a PPI network diagram (Figure 3), in which the larger the “degree value”, the larger the node. The top 10 targets based on their degree values were retained (Table 4): they included tyrosine kinase Src (SRC), epidermal growth factor receptor (EGFR), serine/threonine-protein kinase AKT (AKT1), phosphoinositide-3-kinase regulatory subunit 1 (PIK3R1), tyrosine-protein phosphatase non-receptor type 11 (PTPN11), estrogen receptor 1 (ESR1), androgen receptor (AR), Aldo-keto reductase 1C3 (AKR1C3), tyrosine-protein phosphatase non-receptor type 1 (PTPN1), and cyclin-dependent kinase 1 (CDK1).

### 2.4. GO and KEGG Enrichment Analyses

To further illustrate the biological functions of the 115 intersecting targets, we imported them into the DAVID database for GO and KEGG enrichment analyses. GO examination consisted of three modules: Biological process (BP), Molecular function (MF), and Cellular component (CC). We sorted the top 15 GO items in each module according to the count number (from the largest to the smallest) and then drew the GO function enrichment map (Figure 4). The targets in BP were mainly involved in the oxidation–reduction process, signal transduction, and negative regulation of apoptotic process; the targets in MF mostly took part in protein binding, ATP binding, and zinc ion binding; the targets in CC participated primarily in the cytosol, nucleus, and plasma membrane.

The KEGG pathway analysis returned 74 items, and we selected the top 20 for a visual analysis based on gene count number rank (Figure 5). The analysis revealed that these targets were predominantly enriched in pathways in cancer, proteoglycans in cancer, focal adhesion, insulin resistance, and adherens junction.

### 2.5. Molecular Docking of Vital Active Ingredients and Core Targets

This study selected a portion of the molecular docking results for display in this study (Figure 6). Five vital active ingredients (quercetin, kaempferol, luteolin, naringenin, and sitosterol) and ingredient-related targets (CYP1B1, SRC, ESR2, ABCG2, CYP19A1, AhR, CA4, ALOX5, ABCC1, ESR1, and NOS2) were designated as ligands and receptors, respectively. A binding energy score of the ligand and receptor less than −5 kcal/mol indicates a strong affinity, as reported previously [10].

### 2.6. Validation of RSBDP Treatment Effectiveness and Targets

To further study the therapeutic effect of RSBDP on treating IBD, we constructed the experimental colitis model using 3% DSS and treated it with RSBDP (Figure 7a). Firstly, we found that the body weight of DSS-induced mice started to decrease from the seventh day. After RSBDP treatment on the ninth day, the body weight of the DSS + RSBDP group gradually recovered, while the body weight of the DSS group still decreased (Figure 7b). Afterward, the intestinal tissues from the mice were removed, and the length of their intestines were measured. The intestines of the DSS group were significantly shorter than the DSS + RSBDP group and still showed signs of congestion and inflammation (Figure 7c). Histological analysis further indicated the remarkable attenuation of inflammatory cell infiltration and mucosal damage in DSS + RSBDP (Figure 7d). Moreover, the ki67 expression level was higher in the DSS + RSBDP group than in the DSS and control groups (Figure 7e). Figure 7f showed that three genes related to the intestinal barrier (*ZO-1*, *Occludin*, and *Claudin-1*) were up-regulated in the DSS + RSBDP group but not statistically different from the DSS group. Serum concentrations of IL-1, IL-6, and IL-8 were lower in the DSS + RSBDP group than in the DSS group, among which the level of IL-8 was significantly lower (Figure 7g). Similarly, the protein level of TNF-α was also declined in DSS + RSBDP group (Figure 7i).

In order to further reveal the underlying mechanism of RSBDP in the treatment of IBD, we performed qRT-PCR to quantify the mRNA abundance of the key targets of RSBDP against IBD in Table 3. As expected, these key targets, such as *CA7*, *CPY1B1*, and *PTPN11*, were significantly up-regulated in the DSS + RSBDP group. In particular, the expression level of *CYP1B1* in the DSS + RSBDP group was up-regulated by as high as 9-fold compared to the DSS group (Figure 7h). Finally, we detected the protein levels of key targets (as shown in Table 4) and their downstream targets. Western blotting results indicated that the protein levels of AKR1C1, PI3K, AKT, p-AKT, Bax, and Bcl-2 were significantly up- and down-regulated in the DSS + RSBDP group. The quantitative analysis of western blot is shown in Figure 7j. Altogether, RSBDP affects the development of IBD by pro-apoptosis and remodeling the intestinal epithelial barrier in the DSS-induced IBD model.

## 3. Discussion

IBD is a chronic, relapsing, non-specific inflammatory disease of the intestine and is recognizable in two forms, UC and CD. It is persistent and destructive and can cause a variety of complications, including abscesses, fistulas, bleeding, and colitis-related tumors, and cancers [11]. Therefore, finding effective treatments is a high clinical imperative for patients subjected with IBD. While there is no official name for IBD in Chinese medicine, its clinical features are often identified as “dysentery”, “diarrhea”, and “hemorrhoidal hamorrhage”. RSBDP, a well-known formula in Chinese medicine for dysentery therapy, is currently being used to treat IBD and has yielded decent clinical results [12]. To elucidate the potential molecular mechanism of RSBDP’s treatment of IBD, we quarried the TCMSP database and screened for the main active ingredients of RSBDP against IBD, obtaining a total of 39 active compounds, comprised predominantly of quercetin, kaempferol, luteolin, naringenin, and sitosterol (Top 5). This is not unexpected, as several studies have demonstrated that these active ingredients (Top 5) relieve symptoms of drug-induced colitis and maintain the integrity of the intestinal epithelium [13,14,15,16,17,18]. Remarkably, among the 39 main active ingredients, those with similar effects but available in lower quantities, like tanshinone IIa [19], eriodictyol [20], arachidonate [21], acacetin [22], and nobiletin [17], have also displayed acceptable clinical effects in the colitis model. Therefore, RSBDP possibly exerts its therapeutic influence against IBD through the synergy of multiple ingredients.

It is recognized that IBD is a complex immune disease affected by genetic and environmental factors, and a deeper understanding of the unique role of the intestinal epithelium in its pathogenesis appears key to discovering potential targets for drug therapy [23]. Therefore, genes involved in pathways, such as intestinal barrier integrity [24], adaptive immunity [25], inflammation and fibrosis [23], and inflammasome signaling [26], could all be potential therapeutic targets. Therefore, topological, PPI, GO, and KEGG analyses of key targets could provide some insights into RSBDP’s actions against IBD.

The aryl hydrocarbon receptor (AhR) is one of the potential targets of RSBDP’s treatment of IBD, exerting various regulatory effects through binding to flavonoids or natural drugs and influencing the transduction of downstream signaling pathways [27,28]. Evidence suggests that AhR can induce the regulation of the expression of a series of CYP enzymes [29], pointing to flavonoid-related ingredients of RSBDP’s ability to regulate and maintain the homeostasis of the intestinal epithelium by activating the AhR/CYP pathway. Consistent with these reports, our molecular docking results indicated that kaempferol bound well with AhR (Figure 6f). The qRT-PCR results showed that oral administration of RSBDP significantly promoted the expression of *CYP1B1* (Figure 7h). Particularly, members of the CYP family participate in the activation and suppression of inflammation via the synthesis and breakdown of bioactive mediators (e.g., converting fatty acids into pro- or anti-inflammatory factors) [29,30]. The CYP family is also involved in the synthesis and metabolism of various hormones in organisms, and certain hormones (e.g., estrogen and progesterone) promote the wound healing of intestinal epithelium, alleviate the endoplasmic reticulum stress, reduce the pro-inflammatory factors secretion, and improve intestinal epithelial cell barrier [31].

To further reveal the mechanism of RSBDP against IBD, we integrated and analyzed the results of PPI, GO, and KEGG, focusing on the AKR1C1/PI3K/AKT pathway. Although the gene and protein sequences of mouse AKR1C3 were not available in the NCBI database, a previous study analyzed the homology of human AKR1C3 with mouse AKR1C family, and suggested that the mouseAKR1C1 (also known as AKR1C6 in mouse) and human AKR1C3 has higher homology and the similar functions [32]. Interestingly, several studies pointed out that the decreased levels of AKR1C1/AKR1C3 protein can inhibit the phosphorylation of AKT [33,34]. In this study, oral RSBDP decreased the protein concentration of AKR1C1, PI3K, AKT, and p-AKT, which hinted that RSBDP regulates the intestinal epithelial barrier through inhibition of the AKR1C1/PI3K/AKT pathway in the DSS-induced mice. In addition, research shows that PI3K/AKT signaling pathway plays an essential role in cellular processes and apoptosis [35]. Hence, Bax and Bcl-2, two apoptosis-related proteins, were detected in this study. Some evidence suggested that sodium selenite, melatonin, and dapagliflozin alleviate IBD by anti-apoptosis in experimental IBD model [36,37,38]. Unexpectedly, our study indicated that the protein concentrations of Bax and Bcl-2 were up- and down-regulated after oral RSBDP, respectively. Thus, our results demonstrate that RSBDP displays a unique role in leading to the alleviation of IBD by promoting apoptosis in intestinal epithelial cells. Similarly, several reports suggested that pro-apoptosis may be a new approach to treating colitis [39,40,41].

In this study, we confirmed the positive effect of RSBDP on DSS-induced mice at the molecular level. However, the ingredients of RSBDP are pretty complex, and it would be fascinating to explore the therapeutic effects of critical ingredients of RSBDP in IBD. As the vital natural flavonoid ingredients of RSBDP, quercetin, kaempferol, luteolin, and naringenin have all been demonstrated to resist IBD through anti-inflammatory and an-tioxidant pathways [13,14,15,16,17]. Consistent with these studies, our results confirmed that RSBDP exerts its anti-inflammatory effects (significantly reduces serum IL-1, IL-6, and IL-8 concentrations) through these critical ingredients in the IBD model. Remarkably, the natural flavonols are important ligands for AhR, and the activation of AhR plays a pivotal role in the development of IBD [24,42]. Here, the molecule docking results revealed that kaempferol and AhR had an excellent bonding capability (−6.8 kcal/mol), which hints that the major active ingredients of RSBDP might act as ligands of AhR to activate downstream signaling pathways. Moreover, a few studies demonstrated that quercetin cannot alleviate the symptoms of colitis in AhR^−/−^ mice [13]. Hence, the critical ingredients of RSBDP might combat IBD by activating AhR pathway. A recent study indicated that the aromatic compounds in coffee could promote the expression of *CYP1A1* and *CYP1B1* by activating AhR, which alleviates experimental colitis [43]. Our results also demonstrated that RSBDP promotes *CYP1B1* expression through AhR, reducing inflammation in IBD models. RSBDP also promotes the apoptosis of intestinal epithelial cells via AKR1C1/PI3K/AKT and, thus, against IBD. Although there is no direct evidence that the critical ingredients of RSBDP can alleviates the symptoms of IBD by affecting the concentration of *AKR1C1*, indirect evidence suggests that the addition of kaempferol inhibited the mRNA level of *AKR1C1* and induced apoptosis in non-small cell lung cancer cells [44]. Finally, RSBDP alleviation of IBD may function, but is not limited to, through the five major ingredients (quercetin, kaempferol, luteolin, naringenin, and sitosterol).

## 4. Materials and Methods

### 4.1. Screening for the Main Active Ingredients of RSBDP

Data on the main active compounds of RSBDP were retrieved from the Traditional Chinese Medicine Systems Pharmacology Database and Analysis Platform (TCMSP, https://lsp.nwu.edu.cn/tcmsp.php, accessed on 4 September 2021) [45]. Oral bioavailability (OB) and drug likeness (DL) indices were used to evaluate drug feasibility. In this study, molecules with OB of ≥30% and DL of ≥0.18 were considered as the potential active compounds based on screening parameters. The chemical structural formula of ingredients of RSBDP were obtained from the PubChem database (https://pubchem.ncbi.nlm.nih.gov/, accessed on 6 September 2021) by importing the unique IDs of the retrieved compounds [46].

### 4.2. The Target Prediction of RSBDP Ingredients for IBD Therapy

The collated structural formulae of the main active ingredients were imported into the SwissTargetPrediction database [47] (http://www.swisstargetprediction.ch/, accessed on 15 September 2021) for molecular docking and screened potential targets using a probability threshold ≥0.5. Meanwhile, the keywords “inflammatory bowel disease” were input into GeneCards [48] (https://www.genecards.org/, accessed on 15 September 2021) and OpenTargets [49] (https://www.opentargets.org/, accessed on 15 September 2021) databases to search for IBD-related proteins. The IBD targets obtained from both databases were first merged and then overlapped with the potential targets acquired by prediction from the SwissTargetPrediction database for subsequent analysis.

### 4.3. The Construction and Analysis of a “Drug-Ingredient-Target” Network

We verified the results from Section 2.2, removing ineffective targets and importing effective targets into Cytoscape 3.7.2 to construct a drug-ingredient-target network and perform topological analysis [50].

### 4.4. The Construction and Analysis of the Protein-Protein Interaction Network

The screened 115 targets were introduced into the STRING database [51] (https://string-db.org/, accessed on 15 September 2021) and set the “minimum required interaction score” for the protein-protein interaction network (PPI) analysis to 0.7 [10]. The PPI network was mapped using the Cytoscape software.

### 4.5. Gene Ontology and Kyoto Encyclopedia of Genes and Genomes Signal Pathway Enrichment Analysis

We performed the Kyoto Encyclopedia of Genes and Genomes (KEGG) and Gene Ontology (GO) enrichment analyses on the targets from Section 2.3 using the DAVID database. After downloading and collating the relevant data, we visualized them using the ggplot2 package in the R software.

### 4.6. Molecular Docking Analysis

Molecular docking analyses of the screened key ingredients and their targets were conducted. The 3D molecular structures of the vital active ingredients and their targets were obtained from the Pubchem [46] and PDB [52] databases, respectively, and the Autodock software was employed for the removal of the water molecules of target proteins and generation of PDBQT format files. Molecular docking and importation of the downstream files into the Pymol software to create visual images were carried out using the Autodock Vina program.

### 4.7. Animal Experiment

#### 4.7.1. Drug

In this study, Panax ginseng, Radix bupleuri, Radix Peucedani, Rhizoma Ligustic Chuanxiong, Fructus Aurantii, Rhizoma et Radix Notopterygii, Radix Angelicae Pubescentis, Poria, Radix Platycodonis, Radix Glycyrrhizae, Rhizoma Zingiberis Recens, and Herba Menthae were purchased from the outpatient department of Chengdu University of traditional Chinese medicine (Chengdu, China), and they were conformed to the quality standards of Chinese Pharmacopoeia (2015 edition). Afterward, these herbs were authenticated by Prof. Jin Pei (Department of Pharmacognosy, Chengdu University of traditional Chinese medicine). Finally, all herbs were crushed separately and mix the above herbs according to the ratio 2:4:2:3:2:2:2:2:2:2:1:1, then added 750 mL (1:10 g/v) of pure water and boiled for 20 min. Subsequently, the drug solution was filtered through a 0.45um filter and concentrated to 75 mL (stored at −20 °C).

#### 4.7.2. Animals and Experimental Design

Six-week-old male C57BL/6 mice were maintained with five to six animals per cage and house in specific pathogen-free facility with a 12-h light and 12-h dark cycle at 22 °C.

All mice were randomly divided into three groups, control group (*n* = 5), DSS group (*n* = 6) and RSBDP+DSS group (*n* = 6). 3% dextran sodium sulfate (DSS, MP Biomedicals) was added to drinking water for 9 days. For the next 5 days, 0.308 mL of saline was gavaged daily in the control and DSS groups, and 0.308 mL of drug solution was gavaged daily in the RSBDP + DSS group. All mice were euthanized at the end of the experiment. Tissues or serum were frozen in liquid nitrogen, and colon specimens were fixed in 4% paraformaldehyde.

#### 4.7.3. Histological Analysis

Hematoxylin and eosin (H&E) was used to measure the degree to which the colon had been damaged and inflamed, and the epithelial proliferation was detected using Ki67 (1:500, Servicebio, Wuhan, China).

#### 4.7.4. Enzyme-Linked Immunosorbent Assay

IL-1, IL-6, and IL-8 were performed on serum using the ELISA Kit according to the manufacturer’s instructions (HYCEZMBIO, Wuhan, China).

#### 4.7.5. RNA Extraction and Quantitative Real-Time PCR (qRT-PCR)

RNA extraction was performed using EASYspin Plus Kit (Aidlab, Beijing, China) following the manufacturer’s instructions, and the HiScript Ⅱ qRT SuperMix R223 (Vazyme, Nanjing, China) was used to transcribe total RNA into cDNA. The ChamQ Universal SYBR qPCR Master Mix Q711 (Vazyme, Nanjing, China) was used for qRT-PCR. The primer sequences are listed in Appendix A.

#### 4.7.6. Western Blotting

Tissues were lyzed in RIPA buffer (Biosharp, Hefei, China) containing phenylmethylsulfonyl fluoride (PMSF) and phosphatase inhibitors, and the extracts were centrifuged at 14,000 rpm for 10 min at 4 °C. The 10% SDS-PAGE was used to separate total protein, then transferred onto the polyvinylidene fluoride (PVDF) membrane. Primary antibodies include AKR1C1 (A13004), PI3K (A11526), Bax (A19684), Bcl-2 (A20736), TNF-α (A11534), and β-actin (AC028) were purchased from ABclonal (Wuhan, China); AKT (AF6261) was purchased from Affinity (Cincinnati, OH, USA); p-AKT (80455-1-RR) was purchased from Proteintech (Wuhan, China). Anti-rabbit or -mouse IgG conjugated to horseradish peroxidase was applied after secondary antibody incubation for protein detection.

#### 4.7.7. Statistics

The experimental results are presented as means ± SEM for each group with at least three independent experiments. Data analysis using one-way ANOVA followed by LSD test. The *p* < 0.05 was considered statistically significant.

## 5. Conclusions

In summary, we firstly performed the network pharmacology combined with molecular docking approach to elucidate that the anti-IBD effects and the underlying mechanism of RSBDP, and the key targets of RSBDP against IBD were partially identified by experiments, including CA7, CYP1B1, and PTPN11. In addition, western blotting results revealed multiple functions of RSBDP in the DSS-induced colitis which were exerted by anti-inflammatory and pro-apoptotic activities, relying on the AhR/CYP1B1 and AKR1C1/PI3K/AKT pathways.

## Figures and Tables

**Figure 1 pharmaceuticals-15-01038-f001:**
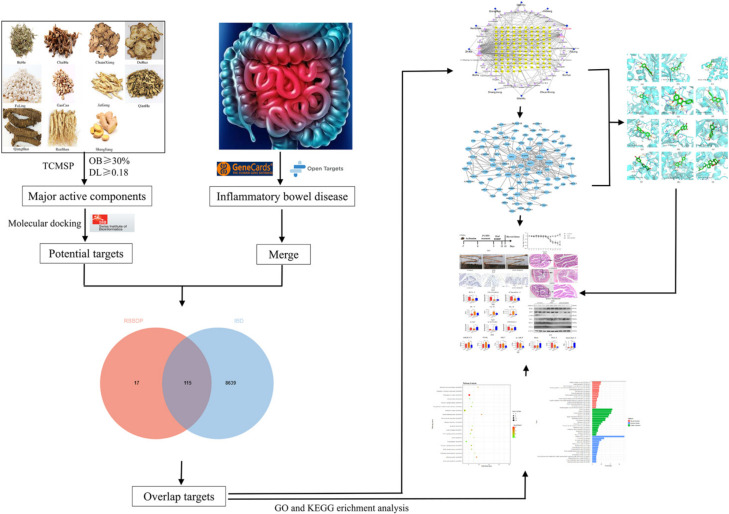
The technology roadmap of network pharmacology in this study.

**Figure 2 pharmaceuticals-15-01038-f002:**
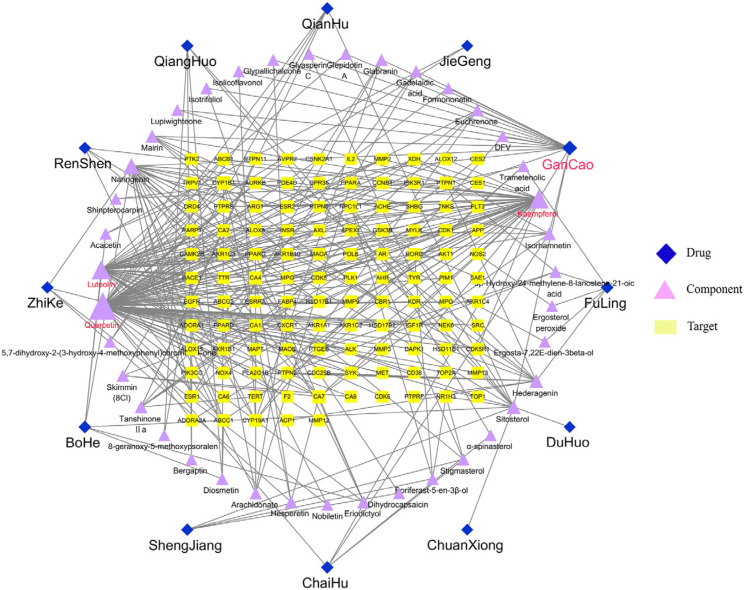
The drug–ingredient–target network for RSBDP in the treatment of IBD. The dark blue diamond the drug; the light purple triangle the pharmaceutical ingredients; the yellow rectangle the potential targets. Node size represents degree value.

**Figure 3 pharmaceuticals-15-01038-f003:**
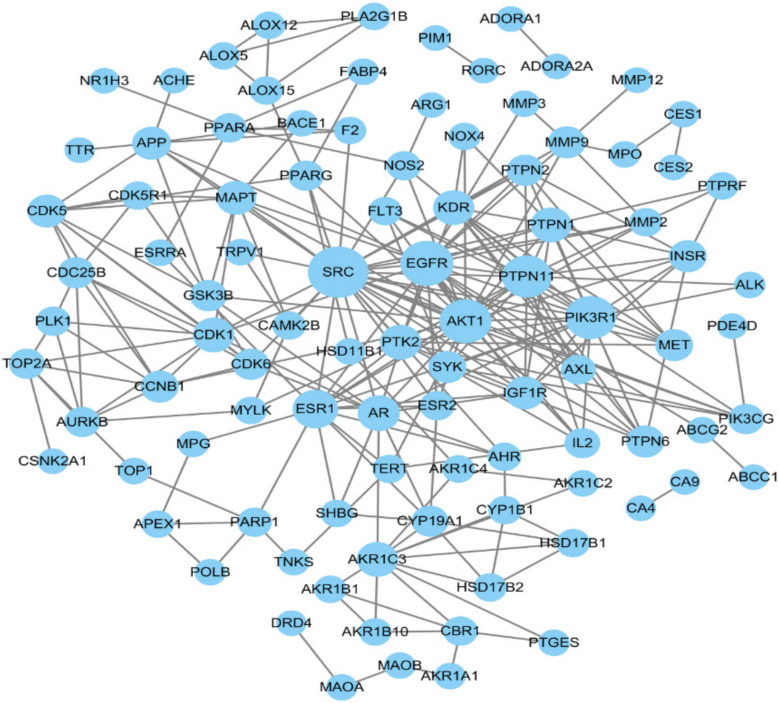
The PPI network of targets. Node size represents degree value.

**Figure 4 pharmaceuticals-15-01038-f004:**
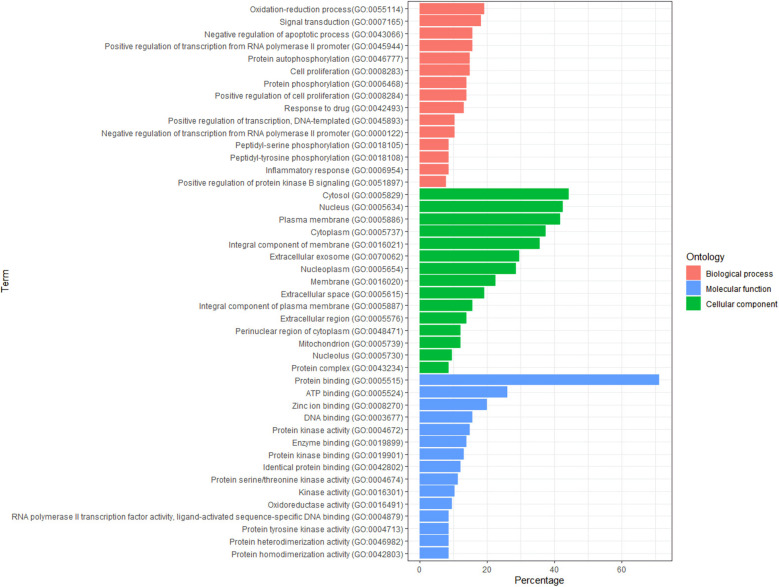
The GO analysis of key targets.

**Figure 5 pharmaceuticals-15-01038-f005:**
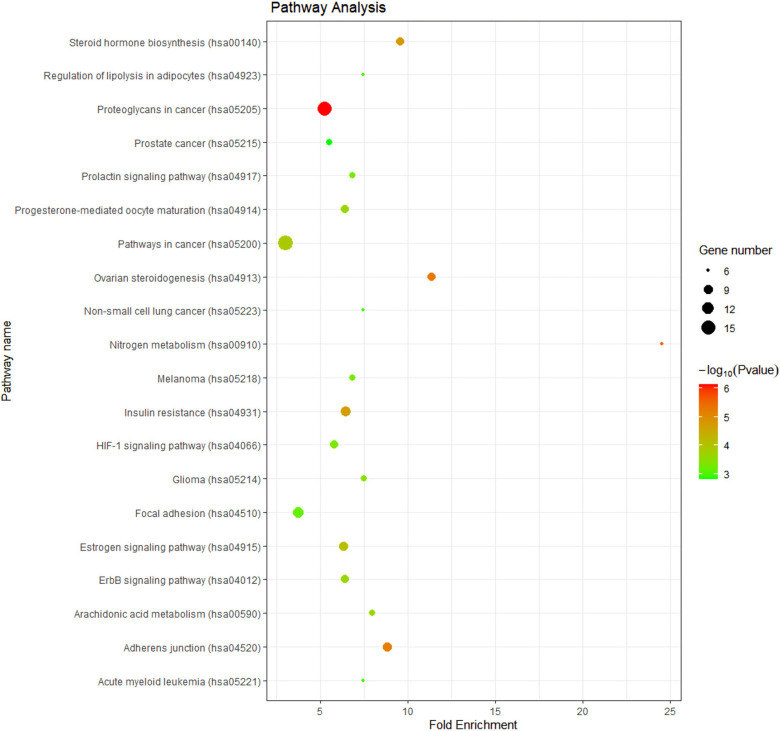
The KEGG pathway analysis of key targets.

**Figure 6 pharmaceuticals-15-01038-f006:**
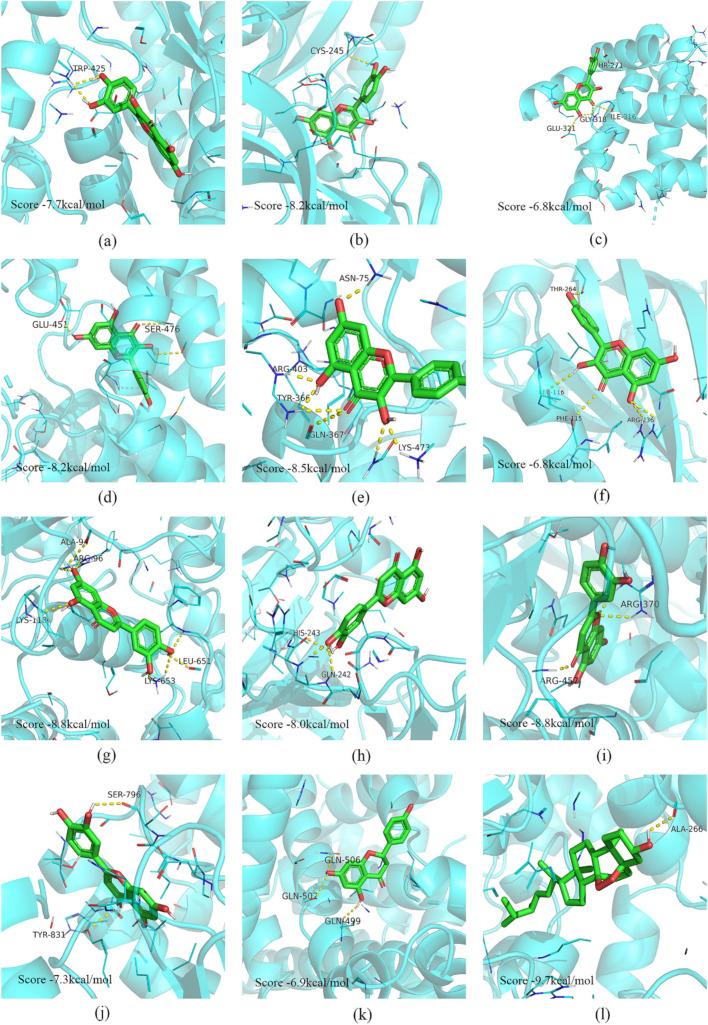
Molecular docking model of vital active ingredients and core targets. (**a**) Quercetin−CYP1B1; (**b**) Quercetin−SRC; (**c**) Quercetin−ESR2; (**d**) Kaempferol−ABCG2; (**e**) Kaempferol−CYP19A1; (**f**) Kaempferol−AhR; (**g**) Luteolin−ABCG2; (**h**) Luteolin−CA4; (**i**) Luteolin−ALOX5; (**j**) Luteolin−ABCC1; (**k**) Naringenin−ESR1; (**l**) Ergosterol peroxide−NOS2.

**Figure 7 pharmaceuticals-15-01038-f007:**
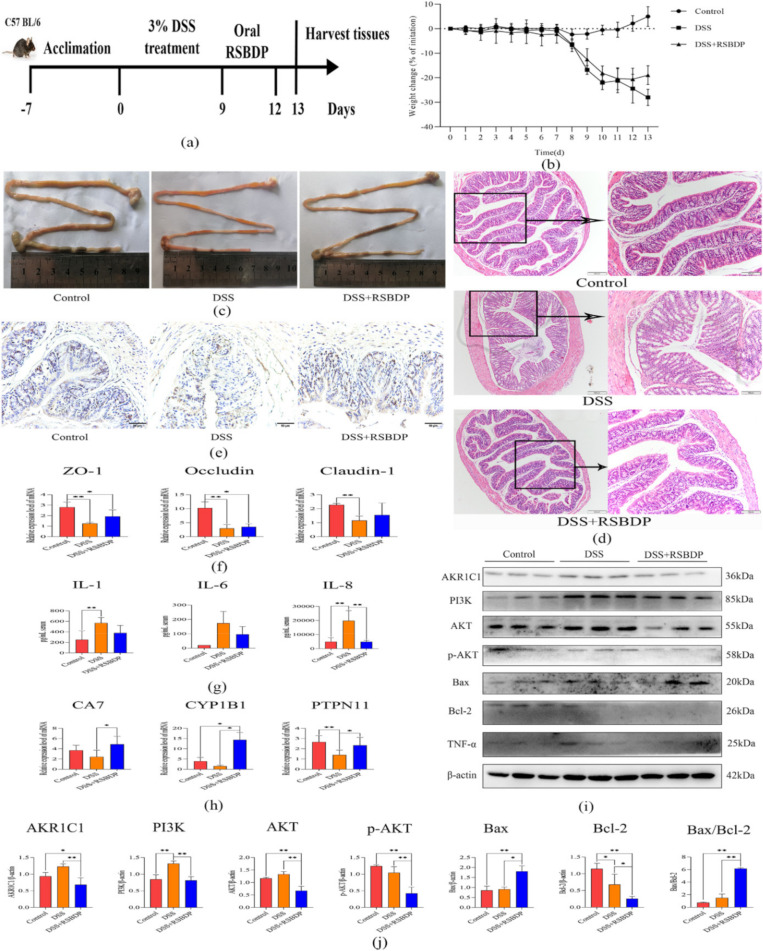
Experimental validation in DSS-induced colitis model. (**a**) An illustration of the mouse model of colitis used in this study. (**b**) Changes in body weight of mice after DSS and RSBDP treatment. (**c**) The intestines from control and IBD mice with RSBDP therapy. (**d**) H&E staining of the colon from control, DSS, and DSS + RSBDP groups. (**e**) The immunohistochemical results of the proliferation-associated maker gene Ki67. (**f**) qRT−PCR results of ZO−1, Occludin, and Claudin−1. (**g**) The concentrations of IL−1, IL−6, and IL−8 in serum. (**h**) qRT−PCR detection of the partial targets in Table 3 and Table 4. (**i**) Western blot analysis of AKR1C1, PI3K, AKT, p−AKT, Bax, Bcl−2, and TNF−α. (**j**) The quantitative analysis of western blot. Data were shown as means ± SEM (*n* = 3–5 per group). * *p* < 0.05, ** *p* < 0.01.

**Table 1 pharmaceuticals-15-01038-t001:** The primary active ingredients of RSBDP.

Drug	Mol ID	Molecule Name	OB (%)	DL
ChaiHu	MOL000354	Isorhamnetin	49.60	0.31
	MOL000422	Kaempferol	41.88	0.24
	MOL000098	Quercetin	46.43	0.28
	MOL000449	Stigmasterol	43.83	0.76
	MOL004718	α-spinasterol	42.98	0.76
ChuanXiong	MOL000359	Sitosterol	36.91	0.75
DuHuo	MOL000358	β-sitosterol	36.91	0.75
FuLing	MOL000287	3β-Hydroxy-24-methylene-8-lanostene-21-oic acid	38.70	0.81
	MOL000282	Ergosta-7,22E-dien-3β-ol	43.51	0.72
	MOL000283	Ergosterol peroxide	40.36	0.81
	MOL000296	Hederagenin	36.91	0.75
	MOL000275	Trametenolic acid	38.71	0.80
GanCao	MOL001792	DFV	32.76	0.18
	MOL004806	Euchrenone	30.29	0.57
	MOL000392	Formononetin	69.67	0.21
	MOL004996	Gadelaidic acid	30.70	0.20
	MOL004910	Glabranin	52.90	0.31
	MOL004828	Glepidotin A	44.72	0.35
	MOL004811	Glyasperin C	45.56	0.40
	MOL004835	Glypallichalcone	61.60	0.19
	MOL004949	Isolicoflavonol	45.17	0.42
	MOL000354	Isorhamnetin	49.60	0.31
	MOL004814	Isotrifoliol	31.94	0.42
	MOL000422	Kaempferol	41.88	0.24
	MOL003656	Lupiwighteone	51.64	0.37
	MOL000211	Mairin	55.38	0.78
	MOL004328	Naringenin	59.29	0.21
	MOL000098	Quercetin	46.43	0.28
	MOL004891	Shinpterocarpin	80.30	0.73
	MOL000359	Sitosterol	36.91	0.75
JieGeng	MOL001689	Acacetin	34.97	0.24
	MOL000006	Luteolin	36.16	0.25
QianHu	MOL005100	5,7-dihydroxy-2-(3-hydroxy-4- methoxyphenyl)chroman-4-one	47.74	0.27
	MOL000358	β-sitosterol	36.91	0.75
	MOL000098	Quercetin	46.43	0.28
	MOL013083	Skimmin (8CI)	38.35	0.32
	MOL007154	tanshinone Ⅱa	49.89	0.40
RenShen	MOL005320	Arachidonate	45.57	0.20
	MOL000358	β-sitosterol	36.91	0.75
	MOL000422	kaempferol	41.88	0.24
	MOL000449	Stigmasterol	43.83	0.76
ZhiKe	MOL000358	β-sitosterol	36.91	0.75
	MOL002341	Hesperetin	70.31	0.27
	MOL004328	Naringenin	61.67	0.52
	MOL005828	Nobiletin	61.67	0.52
BoHe	MOL005190	Eriodictyol	71.79	0.24
ShengJiang	MOL008698	Dihydrocapsaicin	47.07	0.19
	MOL001771	Poriferast-5-en-3β-ol	36.91	0.75

Note: OB: oral bioavailability; DL: drug-likeness.

**Table 2 pharmaceuticals-15-01038-t002:** Key molecules and topological parameters of RSBDP against IBD (Top 5).

Ingredients	Betweenness Centrality	Closeness Centrality	Degree
Quercetin	0.55011787	0.49848943	76
Kaempferol	0.15632436	0.39568345	40
Luteolin	0.1629187	0.3724605	37
Naringenin	0.04758899	0.33199195	23
Sitosterol	0.11861359	0.3674833	13

**Table 3 pharmaceuticals-15-01038-t003:** Topological analysis of key targets of RSBDP against IBD (Top 10).

Targets	Betweenness Centrality	Closeness Centrality	Degree
Cytochrome P450 1B1 (CYP1B1)	0.03810726	0.38461538	11
Carbonic anhydrase 7 (CA7)	0.021592	0.37931034	9
Cytochrome P450 19A1 (CYP19A1)	0.01553419	0.35031847	8
Carbonic anhydrase 4 (CA4)	0.01578918	0.37757437	8
Protein-tyrosine phosphatase 1 (PTPN1)	0.03389732	0.264	6
Estrogen receptor 2 (ESR2)	0.00985945	0.35791757	6
Multidrug resistance-associated protein 1 (ABCC1)	0.01345788	0.37585421	6
ATP-binding cassette sub-family G member 2 (ABCG2)	0.01657	0.37078652	5
Estrogen receptor 1 (ESR1)	0.01515971	0.2519084	4
Cyclin-dependent kinase 1 (CDK1)	0.00317045	0.36423841	4

**Table 4 pharmaceuticals-15-01038-t004:** Topological analysis of the protein-protein interaction of RSBDP against IBD (Top 10).

Targets	Betweenness Centrality	Closeness Centrality	Degree
Tyrosine kinase Src (SRC)	0.27275215	0.48369565	29
Epidermal growth factor receptor (EGFR)	0.1388039	0.44278607	22
Serine/threonine-protein kinase AKT (AKT1)	0.17437856	0.45641026	21
Phosphoinositide-3-kinase regulatory subunit 1 (PIK3R1)	0.05651908	0.41588785	20
Tyrosine-protein phosphatase non-receptor type 11 (PTPN11)	0.0334887	0.38695652	19
Estrogen receptor α (ESR1)	0.15395677	0.41588785	15
Androgen receptor (AR)	0.2042839	0.41784038	12
Aldo-keto reductase 1C3 (AKR1C3)	0.20965192	0.33584906	11
Tyrosine-protein phosphatase non-receptor type 1 (PTPN1)	0.01003103	0.35742972	11
Cyclin-dependent kinase 1 (CDK1)	0.07792777	0.38864629	11

## Data Availability

Data is contained within the article and Appendix A.

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
