# Peer review of "Exploring the Underlying Mechanism of Ren-Shen-Bai-Du Powder for Treating Inflammatory Bowel Disease Based on Network Pharmacology and Molecular Docking"

_pharmaceuticals, 2022, doi:10.3390/ph15091038_

Round 1

Reviewer 1 Report

Major Comments

Discussion: Author should focus on the experimental results of the Colitis model. It would be interesting to see the correlation of experimental results with the selective active ingredients such as quercetin, kaempferol, luteolin, etc. 

Line 305: Mention the parts of plants, and how these are processed before mixing. The authentication of herbal materials also needs to be added to the manuscript. Give the reference for the preparation.  

Minor comments

Line 46: Give the most updated information regarding IBD and the number too. These numbers should reflect the world’s population, not any specific country.

Line 59: For RSBDP composition, mention the parts of each herb.

Figure 1 is not clearly visible. Its quality must be improved. This figure can be used as graphical abstract.

Author Response

Dear Reviewer,

Thank you very much for your excellent comments, which I think will help to improve the quality of our manuscript. Then, I will respond to each of your questions.

Q1: Discussion

Thank you for pointing out the questions that exist in the discussion section of the manuscript. We have added the following to the discussion section:

In this study, we confirmed the positive effect of RSBDP on DSS-induced mice at the molecular level. However, the ingredients of RSBDP are pretty complex, and it would be fascinating to explore the therapeutic effects of critical ingredients of RSBDP in IBD. As the vital natural flavonoid ingredients of RSBDP, quercetin, kaempferol, luteolin, and naringenin have all been demonstrated to resist IBD through anti-inflammatory and an-tioxidant pathways[13-17]. Consistent with these studies, our results confirmed that RSBDP exerts its anti-inflammatory effects (significantly reduces serum IL-1, IL-6, and IL-8 concentrations) through these critical ingredients in the IBD model. Remarkably, the natural flavonols are important ligands for AhR, and the activation of AhR plays a pivotal role in the development of IBD[24,42]. Here, the molecule docking results revealed that kaempferol and AhR had an excellent bonding capability (-6.8 kcal/mol), which hints that the major active ingredients of RSBDP might act as ligands of AhR to activate downstream signaling pathways. Besides, a few studies demonstrated that quercetin can not alleviate the symptoms of colitis in AhR-/- mice[13]. Hence, the critical ingredients of RSBDP might combat IBD by activating AhR pathway. A recent study indicated that the aromatic compounds in coffee could promote the expression of CYP1A1 and CYP1B1 by activating AhR, which alleviates experimental colitis[43]. Our results also demonstrated that RSBDP promotes CYP1B1 expression through AhR, reducing inflammation in IBD models. RSBDP also promotes the apoptosis of intestinal epithelial cells via AKR1C1/PI3K/AKT and thus against IBD. Although there is no direct evidence that the critical ingredients of RSBDP can alleviates the symptoms of IBD by affecting the concentration of AKR1C1, indirect evidence suggests that the addition of kaempferol inhibited the mRNA level of AKR1C1 and induced apoptosis in non-small cell lung cancer cells[44]. Finally, RSBDP alleviates IBD may function, but is not limited to, through the five major ingredients (quercetin, kaempferol, luteolin, naringenin, and sitosterol).

Q2: Line 305

It was an oversight when we wrote the manuscript and neglected to certify the herbs. Thank you very much for your correction! And we have made the following modifications in M&M section:

In this study, Panax ginseng, Radix bupleuri, Radix Peucedani, Rhizoma Ligustic Chuanxiong, Fructus Aurantii, Rhizoma et Radix Notopterygii, Radix Angelicae Pubescentis, Poria, Radix Platycodonis, Radix Glycyrrhizae, Rhizoma Zingiberis Recens, Herba Menthae were purchased from the outpatient department of Chengdu University of traditional Chinese medicine (Chengdu, China), and they were conformed to the quality standards of Chinese Pharmacopoeia (2015 edition). Afterward, these herbs were authenticated by Prof. Jin Pei (Department of Pharmacognosy, Chengdu University of traditional Chinese medicine). Finally, all herbs were crushed separately and mix the above herbs according to the ratio 2 : 4 : 2 : 3 : 2 : 2 : 2 : 2 : 2 : 2 : 1 : 1, then added 750mL (1:10 g/v) of pure water and boiled for 20 minutes. Subsequently, the drug solution was filtered through a 0.45um filter and concentrated to 75 mL (stored at -20℃).

Q3: Line 46

We have modified this sentence as follows and inserted a new reference:

Currently, IBD has become a globalized disease, with five million IBD patients globally and a prevalence of 0.5% in some developed countries.

Reference: Zuo, T.; Kamm, M.A.; Colombel, J.F.; Ng, S.C. Urbanization and the gut microbiota in health and inflammatory bowel disease. Nat Rev Gastroenterol Hepatol 2018, 15, 440-452, doi:10.1038/s41575-018-0003-z

Q4: Line 59

We have added a description of the herb ratio:

The earliest known use of Ren-Shen-Bai-Du Powder (RSBDP) can be traced back to Qian Yi’s “Direct formula for children’s drug syndrome” in the Northern Song Dynasty, which consisted of 12 herbs (weight ratio 2 : 4 : 2 : 3 : 2 : 2 : 2 : 2 : 2 : 2 : 1 : 1).

Q5: Figure 1 is not clearly visible

This is probably because we used the compress command when inserting the image, causing it to be unclear. We re-uploaded the figure. Importantly, thank you very much for your suggestion. We think it is necessary to recreate a graphical abstract.

Finally, thank you again for taking the time out of your busy schedule to review our manuscript. If you are dissatisfied with my response, you are welcome to criticize the correction.

With kind regards,

Prof. Yi Zhou,

Chengdu University of Traditional Chinese Medicine

Reviewer 2 Report

1. The abstract is good however some significant findings must be included. This will allow the readers to follow the paper’s contents easily.

2. The quality of figures is poor, should be improved.

3. I found several grammatical errors. I will advise the authors to thoroughly revise the English language of the manuscript or send it to a native English speaker to improve the language.

4. It will be good if they will add molecular dynamics simulation to confirm these studies.

Author Response

Dear Reviewer,

Thank you very much for your excellent comments, which I think will help to improve the quality of our manuscript. Then, I will respond to each of your questions.

Q1: Abstract

Thank you very much for pointing out the shortcomings of the abstract in our manuscript. We have modified the abstract as follows. Due to the character limit, we avoided extraneous content as much as possible. Of course, our revised abstract may still have shortcomings, and we hope you will continue to correct these weaknesses in order to improve the quality of the article.

Ren-Shen-Bai-Du Powder (RSBDP) is currently used for inflammatory bowel disease (IBD) therapy in China. However, its potential mechanism against IBD remains unknown. In this study, we initially identified potential targets of RSBDP against IBD through network pharmacology analysis and molecular docking. Afterwards, the DSS-induced colitis mice model was employed to assess the effects of RSBDP. The results of network pharmacology indicated that a total of 39 main active ingredients in RSBDP generated 309 pairs of drug-ingredient and ingredient-target correspondences through 115 highly relevant targets of IBD. The primary ingredients (quercetin, kaempferol, luteolin, naringenin, and sitosterol) exerted functions through multiple targets that include CYP1B1, CA4/7, and ESR1/2, etc. GO functional enrichment analysis revealed that the targets related to IBD were significantly enriched in the oxidation-reduction process, protein binding, and cytosol. Per the KEGG pathway analysis, pathways in cancer, adherens junction, and nitrogen metabolism were pivotal in the RSBDP’s treatment of IBD. Additionally, molecular docking demonstrated that a set of active ingredients and their targets displayed good bonding capabilities (e.g., kaempferol and AhR with combined energy < 5 kcal/mol). For the animal experiment, oral RSBDP promoted weight recovery, reduced intestinal inflammation, and decreased serum IL-1, IL-6, IL-8 concentrations in the DSS+RSBDP group. Meanwhile, oral RSBDP significantly up-regulated the mRNA levels of CA7, CPY1B1, and PTPN11, especially the expression level of CYP1B1 in the DSS+RSBDP group was up-regulated by as high as 9-fold compared to the DSS group. Western blot results indicated that the protein levels of AKR1C1, PI3K, AKT, p-AKT, and Bcl-2 were significantly down-regulated, and Bax was significantly up-regulated in the DSS+RSBDP group. Compared to the DSS and control groups, the Bax/Bcl-2 value in the DSS+RSBDP group increased 4-fold and 8-fold respectively, which suggested that oral RSBDP promotes apoptosis of intestinal epithelial cells. In short, this study established quercetin, kaempferol, luteolin, naringenin, and sitosterol as the primary key active ingredients of RSBDP that exert synergistic therapeutic effects against IBD through modulating the AhR/CYP1B1 and AKR1C1/PI3K/AKT pathways.

Q2: The quality of figures is poor

This is probably because we used the compress command when inserting the image, causing it to be unclear. We re-uploaded the figure.

Q3: Grammatical errors

Thank you very much for your careful review of our manuscript, and we found a number of grammatical or spelling errors during the full proofreading of the manuscript. Meanwhile, we have corrected the grammar of the whole text according to your suggestion.

For example. “However, there is no official name for IBD in Chinese medicine. Its clinical features can be classified as “dysentery” and “diarrhea” and are mainly induced by damp-heat and a di-et that damages the spleen and stomach”.

However, our modifications may not be perfect, so we hope you will review them again and make corrections. And we will do our best to improve the grammar of the manuscript.

Q4: Molecular dynamics simulation

Thank you for your very amazing suggestions! Unfortunately, although your suggestion is very good and I strongly agree with it, our lab currently lacks the basis for molecular dynamics simulation. Therefore, we are unable to complete the content as you requested. Of course, we will also study in the future and hope to improve the quality in future research as much as possible.

Finally, thank you again for taking the time out of your busy schedule to review our manuscript. If you are dissatisfied with my response, you are welcome to criticize the correction.

With kind regards,

Prof. Yi Zhou,

Chengdu University of Traditional Chinese Medicine
